# Implications of Hypothalamic Neural Stem Cells on Aging and Obesity-Associated Cardiovascular Diseases

**DOI:** 10.3390/cells12050769

**Published:** 2023-02-28

**Authors:** Bhuvana Plakkot, Ashley Di Agostino, Madhan Subramanian

**Affiliations:** Department of Physiological Sciences, Oklahoma State University, Stillwater, OK 74078, USA

**Keywords:** aging, cardiovascular conditions, hypothalamus, neural stem cells, neuroinflammation, obesity

## Abstract

The hypothalamus, one of the major regulatory centers in the brain, controls various homeostatic processes, and hypothalamic neural stem cells (htNSCs) have been observed to interfere with hypothalamic mechanisms regulating aging. NSCs play a pivotal role in the repair and regeneration of brain cells during neurodegenerative diseases and rejuvenate the brain tissue microenvironment. The hypothalamus was recently observed to be involved in neuroinflammation mediated by cellular senescence. Cellular senescence, or systemic aging, is characterized by a progressive irreversible state of cell cycle arrest that causes physiological dysregulation in the body and it is evident in many neuroinflammatory conditions, including obesity. Upregulation of neuroinflammation and oxidative stress due to senescence has the potential to alter the functioning of NSCs. Various studies have substantiated the chances of obesity inducing accelerated aging. Therefore, it is essential to explore the potential effects of htNSC dysregulation in obesity and underlying pathways to develop strategies to address obesity-induced comorbidities associated with brain aging. This review will summarize hypothalamic neurogenesis associated with obesity and prospective NSC-based regenerative therapy for the treatment of obesity-induced cardiovascular conditions.

## 1. Introduction

Neural stem cells (NSCs) in an adult brain are responsible for neurogenesis and regeneration of brain functions. The two primary NSC reservoirs (neurogenic niches) in an adult mammalian brain are the sub-ventricular zone (SVZ) of the lateral ventricles and the hippocampal dentate gyrus (DG) [1,2,3]. In recent times, a third NSC pool, hypothalamic neural stem cells (htNSCs), were discovered [4,5,6]. The htNSC population is sensitive to variations in nutrient intake and signaling. An increase in neurogenesis in the hypothalamus was observed upon acutely feeding a high fat diet (HFD) [7], whereas a reduced neurogenesis in the hypothalamus was noticed as a result of chronic HFD feeding [8], and ‘inflammation’ was suggested as a major factor in causing such pronounced changes in neurogenesis. Upon htNSCs culturing, we observed a significant increase in htNSCs after eight months in HFD-fed C57BL/6J male adult mice compared to the chow-fed controls (unpublished).

Cellular senescence is an irreversible growth arrest in proliferating cells, which has been implicated in several neurodegenerative diseases [9,10]. During the process of senescence, the NSCs lose their ability to proliferate and generate neurons [11,12,13,14]. Supplementing mono-unsaturated fatty acids, such as oleic acid, in the diet caused lipid droplets to develop in ependymal cells and contributed to a decrease in neurogenesis in SVZ in the Alzheimer’s disease mouse model, 3xTg-AD [15]. Likewise in obesity, SVZ showed an increase in senescent glial cells carrying excessive fat deposits, and genetically ablating these senescent glial cells restored neurogenesis [16]. Thus, modifying the lipid content in the diet can replenish the old neurogenic pool. In this review, we will summarize hypothalamic neurogenesis associated with obesity and aging and explore the possibilities of NSC-based regenerative therapy to treat obesity-induced cardiovascular conditions.

## 2. HtNSCs and Obesity

NSCs are multipotent and they generate neurons, oligodendrocytes, and glia in the nervous system [17]. Varied levels of neural inflammation are observed in many neurological disorders or neurodegenerative diseases in human beings [18,19]. Their progression involves mediators of inflammation that are synthesized and secreted by various CNS cells, such as astrocytes, microglia, and oligodendrocytes [20]. Both beneficial and detrimental effects are observed in inflammatory conditions, which makes it unclear to specify the exact role of inflammation on NSCs. Certain pathways, after long term activation, cause energy imbalance, abnormal nutrient metabolism, restricted neurogenesis, proliferation, and differentiation of neural stem cells leading to metabolic and cognitive abnormalities. In the hypothalamus, the medio-basal hypothalamus (MBH) and the 3rd ventricle wall are observed to be the NSC niches [8]. Some studies state that mainly adult NSCs are observed in the MBH [7,21]. The MBH is a predominant region for physiological homeostasis of the entire body. Many neural progenitors or specialized ependymal cells that line the 3rd ventricle are observed to be glia-like tanycytes. They send processes to the arcuate nucleus and ventro-medial nucleus of the hypothalamus. Functionally these tanycytes are observed to be glucosensitive, reacting to metabolic stimulation and signal variations caused by feeding and energy balance [4,7,22]. Properties of tanycytes include ATP release, purinergic P2Y1 receptors, ectonucleoside triphosphate diphosphohydrolase 2 (NTPDase2) expression [23], and reacting to the activation of these receptors by the means of intense Ca^2+^ waves [24]. This is similar to the signaling mechanisms in stem cells. Expression of doublecortin-like [25] proteins, nestin [26,27,28] and vimentin [29,30,31], linked to neural precursor cells are observed in humans and rodent tanycytes. The expression of Sox2 [7,8], a nuclear transcription factor and NSC marker, is found in a few of the tanycytes, especially in the subventricular zone and dentate gyrus. In adult mice, it is mainly expressed in a group of cells in the MBH, particularly within the hypothalamic third-ventricle wall [8]. However, a few studies have shown rare occurrences of proliferating neurogenic progenitors in the human dentate gyrus [32,33]. One of the studies also observed human paralaminar nuclei of the amygdala showing persistence of immature excitatory neurons for decades [34]. Thus, the possibility of observing immature non-proliferative hypothalamic neurons cannot be denied and future studies focusing on confirming their ability to proliferate and differentiate could possibly reveal their normal functionality.

The MBH regulates body weight, feeding, and glucose balance via melanocortin signals based in the arcuate nucleus (ARC), mainly via orexigenic agouti-related peptide (AGRP) neurons and anorexigenic proopiomelanocortin (POMC) neurons [35,36,37,38]. Leptin and insulin, which vary with different fat mass conditions and feeding patterns, affect these two neurons and the process is crucial for body weight homeostasis [36,39,40,41]. The studies also showed decrease in responsiveness to leptin and insulin by these neurons upon chronic feeding of a high-fat diet (HFD), resulting in type-2 diabetes (T2D) and HFD-induced obesity. A 10% loss in POMC neurons was observed in the hypothalamus upon long term HFD feeding [8,21,42]. Neural precursors giving rise to different neurons were observed to have POMC gene expression [43]. Considering these data and mechanisms, there is evidence of dysregulation of neurogenesis in the hypothalamus of obese subjects. Based on many recent studies, neurogenesis has been observed in adult rodents [7,22,44,45] and htNSCs in adult MBH contribute to the regulation of metabolic physiology [8]. Hence, future studies could be focused on developing htNSCs as a treatment regimen for obesity and its related disorders, such as diabetes.

## 3. HtNSCs and Inflammation

Microglia are brain-resident macrophages that contribute to reduced neurogenesis in aging and play a predominant role in the inflammatory response [46]. Through microglia sorting studies, we observed a significant elevation of activated microglia in the hypothalamus of four-month HFD-fed young adult male mice compared to the chow-fed controls (unpublished). Activated microglia have the potential to release proinflammatory cytokines that can be harmful to NSCs, neurons and other glial cells. Among the complex neural immune reactions in adult NSCs, inflammatory cytokines are observed to majorly affect differentiation, proliferation, migration, and survival [47]. Inhibition of neurogenesis is achieved by pro-inflammatory cytokines whereas an increase in neurogenesis is observed by anti-inflammatory cytokines [48]. The gene expression studies in our lab revealed a significant increase in proinflammatory markers, such as IL1β, MCP1, and TNFα, in the whole hypothalamus of middle-aged, eight-month HFD-fed male mice compared to controls (unpublished). However, an anti-inflammatory cytokine, such as the transforming growth factor-beta (TGFβ), can enhance endothelial cells of adult NSC during aging [49,50]. In addition to these, a chemokine, CCL11, was observed to be increased in aged mice, both in blood and cerebrospinal fluid (CSF), which further caused a decline in neurogenesis leading to cognitive function impairment [51]. Exercise and restriction of calories can cause variations in systemic factors, and hence, act as adult NSC function modulators [52].

Upon over-nutrition, the IκB kinase-β/nuclear transcription factor NF-κB (IKKb/NF-κB) pathway, that plays a crucial role in many physiological processes, gets activated; this can cause SOCS3, a suppressor of cytokine signaling-3 gene upregulation in the hypothalamus, to inhibit insulin and leptin signaling, leading to resistance [53]. Studies have confirmed that, in the neurons of the hypothalamus in mice, SOCS3 knockout leads to an improvement in central leptin signaling and reduced obesity [54,55,56]. Similar effects were observed in central IKKb knockout mice and, in the MBH, SOCS3 overexpression decreased the neural IKKb inhibition effect on obesity reduction [53]. Like SOCS3, protein tyrosine phosphatase 1B (PTP1B) causes inhibition of leptin and insulin signaling and was observed to have a role in the IKKb/NF-kB inflammatory pathway. PTP1B expression in the hypothalamus can be increased by TNF-a by activating the IKKb/NF-kB pathway, mainly by being a transcriptional target [57]. Inhibition of PTP1B in neurons resolved leptin resistance, glucose disorders, and obesity induced by over-nutrition [58,59,60]. It is assumed that neural PTP1B may form a link with metabolic disease pathways and neurodegenerative diseases as it had an effect on genetic mouse models of Alzheimer’s disease [61]. In the forebrain, degeneration of GABAergic interneurons was mediated by an overproduction of the cytokine interleukin-6 in diabetes and obesity, which leads to NF-kB activation and release of neurotoxic inflammatory products [62]. Therefore, alleviating chronic diet-induced neuroinflammation by exploring the pathways associated with the metabolic control function of htNSC and identifying their therapeutic potential is essential.

## 4. Nrf2, an Important Transcription Factor Affecting NSC Populations in Obesity

Various factors affect NSC populations in obesity, including hormonal factors, transcription factors, inflammatory factors such as cytokines and chemokines, epigenetic changes and chromatin stability, oxidative stress, DNA damage, hyperlipidemia/hyperglycemia, etc. Nuclear factor E2-related factor 2 (Nrf2) is a major transcription factor that regulates basal and induced expression of antioxidant response element genes in response to oxidative stress. Functions of Nrf2 also include stem cell survival, apoptosis, autophagy, mitochondrial biogenesis, and many more, in addition to aging processes [63,64,65,66,67]. Studies in our lab observed an elevated expression of Nrf2 in the hypothalamus of adult obese male mice, along with a significant increase in htNSCs (unpublished). In a previous study, increased oxidation, or reactive oxygen species in adult mouse NSCs, promoted their ability to generate neurons and proliferate [68]. Self-renewal of stem cells was observed to be regulated by Nrf2, along with differentiation initiation with the support of epigenetic factors and transcription regulators [69]. Nrf2 expression and transcriptional activity steadily increased during the induced oluripotent stem cells (iPSC) differentiation process that peaked in later stages [70]. Restoration of age-related loss of hippocampal function was evidenced by transplanting Nrf2-overexpressing young NSCs [71], indicating the critical role of Nrf2 in mediating NSC/neural progenitor cell (NPC)- dependent neurogenesis in aging. Redox homeostasis by Nrf2 critically mediates the differentiation ability of different stem cell types to survive oxidative stress, which could gradually reduce during aging [69]. Thus, obtaining insight into one of the main transcription factors, Nrf2, that can resist oxidative stress, could provide fundamental knowledge about changes in htNSCs during neuroinflammation and lead to development of an associated therapeutic strategy.

## 5. HtNSCs and Aging

A continuous decline in physiological integrity is observed during aging. Characteristic intertwining factors that contribute to the complex aging process include deregulated nutrient sensing, cellular senescence, epigenetic alterations, genomic instability, loss of proteostasis, mitochondrial dysfunction, telomere attrition, change in intercellular communication, and exhaustion of stem cells [72,73].

It has been observed in various research that the hypothalamus is particularly important in aging [74,75,76,77] but the underlying cellular mechanism is not known in depth. The IκB kinase-β (IKKβ) pro-inflammatory axis in the hypothalamus and its downstream nuclear transcription factor, NF-κB, (IKKβ/NF-κB signaling) is over-stimulated in over-nutrition or aging [77,78,79]. Systemic aging is directed by the hypothalamic IKKβ/NF-κB pathway via inflammatory crosstalk between neurons and microglia by inhibiting gonadotropin-releasing hormone (GnRH) production, and so counteracting inflammation or GnRH therapy could partly regress degenerative signs of aging [77]. Maternal inflammation has been observed to cause reduced ventricular cell proliferation in developing fetal mouse brain [80]. In young mice, a high number of cells co-expressing Sox2 and the polycomb complex protein, Bmi-1, a nuclear protein [81] that is vital for self-renewal of NSCs and hematopoietic stem cells [82], were observed in the third-ventricle wall, whereas the ones in the MBH were found to be sparse. However, a gradual decrease in these cells was observed as age increased, which was initiated in the ventral region of 3rd ventricle wall within the MBH in 11–16-month-old mice and was totally lost in 22-months-and-older ones. Thus, various studies that aim to evaluate the exact time required to intervene in an inflammatory condition/pathway in the brain will provide more understanding upon which to formulate therapeutic clinical strategies for different neural stem cell niches.

Senescent glial cell accumulation is observed in proximity to the lateral ventricles along with excessive fat deposition within them. Upon removal of senescent cells from HFD or obese mice deficient in leptin receptors, neurogenesis being restored and a decline in anxiety-related behavior was observed [16]. Hence from subsequent studies, they concluded that the topmost contributors to obesity-induced anxiety are senescent cells. Therefore, senolytic drugs have opened a novel therapeutic pathway to treat neuropsychiatric disorders.

Alterations in mitochondrial structure and function may cause deleterious effects in adult NSC, which could drive the aging process [83]. Abnormal toxic by-product accumulation, including of reactive oxygen species (ROS), accompanies this event [84]. SOD2, an antioxidant enzyme that is regulated by FoxO3, a transcription factor associated with longevity [85], protects adult NSCs in mice [86]. An increased level of ROS and a decrease in the potential for self-renewal of adult NSCs was observed in mice that were deficient in FoxO1, 3, and 4 [87]. Other dysfunctions of mitochondria that contribute to the aging of NSCs include mitochondrial protein oxidation, variations in mitochondrial membrane composition, and abnormal mitophagy [83,88,89].

During mammalian NSC division, protein segregation is affected by age, mainly by means of diffusion barrier alteration. The stem cells are kept free of damage by the diffusion barrier that facilitates asymmetric segregation of damaged proteins among daughter cells [90]. Like yeast, efficient compartmentalization of cellular damage is achieved in young rodent NSCs and that can protect these proliferative cells. As age advances, this efficiency is reduced, causing aged NSCs to be exposed to excessive cell damage [90].

A mitochondrial function regulator, hypoxia-inducible factor-1α (HIF-1α), is essential for the maintenance of adult NSCs in their hypoxic niches. HIF-1α plays a major part in cell adaptation under hypoxia by inducing transcriptional responses. Thus, for proper adult NSC proliferation and subsequent differentiation, oxygen availability is critically important [91]. An abnormal oxygen-sensing pathway may be responsible for the neurogenic decline in aging [92]. Thus, the use of anti-inflammatory agents along with senolytic and associated htNSC therapy have the potential to strategically counteract diet-induced chronic neuroinflammation and aging. This could possibly pave way to new therapeutic regimens in obesity-induced cardiovascular conditions.

## 6. Molecular Pathways Associated with NSC Inflammation and Aging

Certain nutrient-sensing mechanisms that can be associated with aging have been considered modifiers of adult NSCs. Adult NSC proliferation and differentiation can be stimulated by insulin-like growth factor 1 (IGF-1) [93], and a reduced IGF-1 level has been associated with cognitive aging [94]. However, lifelong IGF-1 exposure may be the reason for an age-related reduction in adult neurogenesis [95].

An important metabolic regulation coordinator is the mammalian target of rapamycin (mTOR), which has two types, viz., mTORC1 and mTORC2 [96,97]. Regulation of body weight and feeding behavior is primarily controlled by mTOR1 using ghrelin and leptin signaling, in addition to control of gluconeogenesis and adipogenesis peripherally in many tissues [97]. Size, morphology, and neuronal cell numbers are controlled by mTORC2, along with energy balance regulation in the hypothalamus. In POMC neurons in aged mice, an elevation in mTOR activity was observed [98], which can indirectly lead to POMC neuronal soma enlargement and a decline in the projection of neurites to the paraventricular nucleus (PVN), which causes age-dependent obesity [99]. It has been observed that, upon intracerebral injection, rapamycin causes mTOR inhibition which further leads to neurite projection and neuronal excitability in POMC, establishing a decline in body weight and food consumption; hence, age acceleration is achieved by the mTOR pathway. Therefore, to delay aging and improve the lifespan, this pathway can be considered a potential target for therapeutic intervention.

As previously discussed, during aging, a decrease in htNSCs was observed [81]. In addition, mice models with gene silencing mediating Bmi1+ depletion in stem cells showed a significant reduction in cognition, sociality, muscle endurance, coordination, and spatial memory. In other mice models, a decline in lifespan was observed in Sox2+ stem cell-depleted animals. Hence, replenishing new htNSC from a newborn mouse into the MBH of a middle-aged mouse could enhance the lifespan and delay age-associated physiological decline [81]. Exogenous implantation of stem cells into the hypothalamus caused secretion of microRNA-containing exosomes, which delayed physiological deficits in aging. Suppression of NF-kB activation was achieved in neurons due to these microRNAs, and GnRH secretion was also restored [81]. As a result, during aging, htNSC loss might cause systemic physiological changes due to underlying inflammation.

Through Wingless-related integration site (Wnt)-mediated signaling by astrocytes, adult NSC expansion is induced in a paracrine manner [100]. As age increases, Wnt3 expression reduces in astrocytes, which causes further neurogenic decline [101]. Expression of survivin is decreased in adult NSCs due to an age-associated decline in Wnt-mediated signaling in the astrocytes that leads to a quiescent phase in adult NSCs [102]. Release of the Wnt inhibitor, DKK1, from astrocytes is increased in NSC niches during aging, which decreases neurogenesis [103]. Extracellular matrix composition, mechanical properties, and arrangement have a role in adult NSC function, which varies with injury, disease, and aging [104,105].

A high fat mass expression and obesity associated gene (FTO) was observed in adult NSCs [106]. A smaller number of BrdU+ and Ki67+ cells was also observed during FTO loss, showing a decline in adult NSC and reduced proliferating capacity, along with a decline in glial and neuronal differentiation, making adult NSC less multipotent. In addition, in adult mice, FTO loss was observed to decrease adult NSC proliferation and caused inhibition of neuronal differentiation in both SGZ and SVZ regions. Thus, adult NSC activity modulation is achieved by FTO through m^6^A modification regulation of selective transcripts that can indirectly affect the gene expression [107,108,109].

Alteration of important signaling for neurodevelopment is observed when a change in nutrient or neurotrophic environment is observed. Brain abnormalities and decreased brain weight, along with altered glial and neuronal protein expression is observed in mice that have a paucity of leptin signaling and varied expression of neuronal and glial proteins [110]. Elevated proliferation and decline in neural stem/progenitor cells (NPC) are observed in adult rats with type II diabetes [111] and in rats showing hyperglycemia. NPCs do not respond to growth factors and form neurospheres (NS) that are smaller in size. In addition, a decline in neurogenesis is observed in type I diabetic mice or rats treated with streptozotocin [112]. Along with anorexigenic response signaling, during fetal life, insulin and leptin help in neuronal development and their neurotrophic effects are mediated by the MAPK (ERK/MAPK) pathway that resulted in phosphorylation of ERK1/2 [113]. Significant neuronal differentiation was induced by leptin in differentiation conditions along with elevated early and late neuronal marker expression [114]; whereas the late neuronal marker neuronal nuclei (NeuN) showed no significant increase and a normal elevation in early neuronal markers, such as doublecortin (DCX) and neuron-specific class III β tubulin (Tuj1), were observed upon insulin exposure [114]. According to these studies, it was concluded that maternal diabetes and differential exposure of the fetus to insulin and leptin could result in reduced growth or macrosomia that could have a significant effect on the development of a fetal brain.

## 7. The Hypothalamus and the Sympathoexcitatory Effect

In various studies of the effects of leptin on the hypothalamus, it has been observed that α-melanocyte-stimulating hormone (α-MSH) or melanotan II (agonist of MC3/4R (MTII)), upon intracerebroventricular (ICV) administration, enhanced sympathetic nerve activity (SNA), however agouti-related protein or MC3/4R broad brain inhibition with ICV SHU9119 blocked leptin’s sympathoexcitatory effect [115,116]. This is based on the understanding that α-MSH and glutamate are two major excitatory signals to the PVN, a cardiogenic center in the hypothalamus (see Figure 1), that can mediate leptin’s sympathoexcitatory effects. POMC neurons synthesize and release α-MSH. These neurons are in arcuate nucleus (ArcN), which projects to various sites in the hypothalamus, including the PVN [117,118,119], and regulates autonomic activity; however, the role of PVN MC3/4 is ambiguous. Glutamatergic signals are received by the PVN from various regions that include the dorsal medial hypothalamus, ventral medial hypothalamus, lateral hypothalamus, and ArcN, wherein elevated SNA is observed due to the action of leptin [120]. A small group of POMC neurons in the ArcN also expresses the glutamate vesicular transporter (VGLUT-2) [121]. PVN glutamate receptors blockade decreases the ArcN’s non-specific chemical stimulation-mediated sympathoexcitatory effects [122,123]. Along with this excitatory signaling, inhibitory neurons, such as neuropeptide Y (NPY) neurons of the ArcN, are projected into the PVN [122,123,124]. NPY neurons are inhibited by leptin in the ArcN [125,126] and PVN neuron firing is inhibited by NPY, which gets stimulated by α-MSH or plasma leptin elevation.

By various studies it has been identified that elevated SNA is observed in obesity, especially in the kidney and hindlimb, for which a leptin increase and hypothalamic melanocortin activity elevation are predominant activities [127]. In mice and rats, expression of NPY in the ArcN/PVN is reduced by diet-induced obesity or, in the ArcN, by NPY mRNA levels [128,129,130]. In the PVN region, obesity-prone rats that were inbred showed a reduction in agouti-related protein/NPY processes [131]. Tonic NPY inhibition decline is essential for leptin-induced sympathetic outflow driven via PVN MC3/4R. It is inferred from this that obesity plays a role in SNA inhibition and it is due to tonic activity of NPY, which further reveals an elevated α-MSH excitation [132]. htNSCs are predominantly found adjacent to the PVN of the hypothalamus (See Figure 2) lining the 3rd ventricle [133]. Based on these studies, there is a need for detailed investigation into the link between the variation in NSC levels associated with different conditions, such as age, diet etc., and sympathoexcitatory activity.

## 8. Time-Restricted Feeding and Its Effect on NSCs

Reduced energy consumption without any effect on nutritional value is characteristic of dietary restriction (DR). It can be alternatively described as caloric restriction (CR) and, in a broader way, termed as periodic fasting, short-term starvation, intermittent fasting (IF), and fasting-mimetic diets [134]. In maintaining proper health and physiology, a crucial role is played by the type and amount of diet [135]. Adult stem cells are important for tissue regeneration and homeostasis and these stem cells can differentiate and self-renew into specialized cell types. Dietary changes, environment, and nutrient variation influence the stem cells via function alteration. In various studies, a positive effect was observed in stem cells when calories were restricted, especially an increase in the function of intestine and skeletal muscle stem cells, in addition to an elevated quiescence of hematopoietic stem cells (HSCs). In addition, time-restricted feeding has been shown to protect neuronal stem cells, intestinal stem cells, and HSCs from injury, especially stroke and neurodegenerative diseases in the brain [136,137,138]. HFD impairs neurogenesis and hematopoiesis, and it can create opportunities for tumorigenesis.

Characteristic changes in metabolic pathways in the brain are achieved by IF, mainly by ketogenic amino acid and fatty acid breakdown and an elevation in stress resistance [139,140]. A neuroprotective effect can be achieved via IF by activation of many signaling pathways [141]. IF in rodents has shown an increase in long-term potentiation (LTP) at synapses in the hippocampus and an increase in hippocampal neurogenesis [138] in comparison with animals with a sedentary lifestyle that are fed ad libitum (AL) diet. BrdU-labeled cell number in the dentate gyrus was elevated in the mice that were intermittently fasted [138]. They also used Ki67 as a marker to evaluate cell proliferation by identifying an increase in dentate gyrus Ki67-labeled cells in mice that were fed with an IF diet. Mice subjected to IF for three months showed an elevated level of hippocampal nestin and NeuN (protein markers for precursor/neuronal stem cells), and also PSD95 (a scaffolding protein that is a potent regulator of synaptic strength) compared to AL mice [141], which demonstrated an increase in hippocampal neurogenesis and a strengthening of synaptic connections after IF. The researchers also showed that a pathway essential for neural stem cell maintenance in the mammalian brain [142], the Notch 1 signaling pathway, was shown to become activated mainly by upregulation of full-length Notch 1, Notch intracellular domain (NICD1), and transcription factor HES5 (involved in the formation of neurospheres) after IF.

The stress resistance ability of brain cells is activated by IF by causing various changes in brain metabolic pathways [140]. The changes in metabolic pathways during IF can be injurious to the brain and through activation of the brain-derived neurotrophic factor (BDNF) signaling pathway, a neuroprotective state is achieved. Downstream transcription factor activation that helps in energy balance and neurogenesis is made possible by BDNF, and one such transcription factor is cAMP response element-binding protein (CREB). To differentiate stem cells into matured neurons, collaboration between the Notch signaling pathway and the CREB and BDNF signaling pathways is essential [143,144,145]. An increase in BDNF and p-CREB expression has been seen in IF compared to AL animals [141].

Without leading to malnutrition, CR is a 20–40% reduction in intake of calories. It is known to cause life-span increase, prolonged onset of diseases that are age related, and decrease in the incidence of cancer in different tissues and organisms [146,147,148,149]. The link between CR and longevity is under the influence of the downregulation of major nutrient sensing pathways, including those of insulin or IGF-1, and signaling by mTOR [84,147,150]. Very few studies have been documented on the positive and negative effects of CR on NSCs.

Two-days-a-week fasting or alternate-days fasting (IF) in animals have been shown to decrease clinical symptoms caused by age-related neuronal diseases such as Alzheimer’s disease, and the animals that were fasted also perform better after stroke, which is an acute injury [151]. After three weeks of a three-month period of IF, an elevated NSC proliferation in the rats and mice dentate gyrus was observed [152,153]. An elevated BDNF was associated with these positive effects. However, various studies showed that neuronal survival ability was altered by fasting rather than induction of NSC proliferation. In the dentate gyrus of mice, an increase in neuron and glia numbers was observed within 72 h of feeding a fasting-mimicking diet (FMD), along with a reduced IGF-1/PKA signaling [152,154]. In addition, an increase in mesenchymal stem and progenitor cell number and proliferation were observed on FMD repeated feeding in aged animals, and in aged mice; rebalanced output from HSCs and progenitors were also observed [154,155]. Therefore, time restricting feeding can be a neuroprotective strategy for replenishing lost NSCs in chronic neuroinflammatory conditions.

## 9. Exosomes from the HtNSCs

HtNSCs have a distinct endocrine function, to release excessive amounts of microRNAs (miRNAs)-containing exosomes [81]. In addtion, they have certain long non-coding RNAs (lncRNAs) that possess the ability to maintain pluripotency and embryonic stem cell neurogenesis [156], self-renewal of cancer stem cells [157], and reprogramming of pluripotent stem cells [158]. LncRNAs may play a unique role in determining the fate of these stem cells in cellular senescence regulation [159]. An abundant lncRNA, Hnscr in the htNSCs of young mice, drastically reduces as the mice age [159]. Hnscr regulates htNSC senescence and mouse aging by binding to YB-1, a multi-functional protein [160] that controls protein translation [161], and also regulates DNA repair [162], protecting it from protein degradation and ubiquitination. YB-1 acts as a repressor of transcription, inhibiting p16^INK4A^ expression in htNSCs [159,163], and hence could be targeted to modulate senescence in htNSC. According to [159], TF2A treatment, isomeric theaflavin monomer, and a black tea derivative [164], improved YB-1 stability, diminished htNSC senescence, and decreased the level of aging related physiological downturn in mice.

By various studies it has been observed that htNSC loss causes systemic aging within a short time and the exosomal miRNAs secreted by these cells (See Figure 3) mediate anti-aging properties [81]. Aging can be correlated with modulation of some gene expressions by certain non-coding RNAs. In aged adult NSC, heterochronic micro-RNA let-7b upregulation is observed. Repression of Hmga2, a high mobility group transcriptional regulator, is observed upon let-7b overexpression, which indirectly potentiates p16^lnk4a^ (an inhibitor of cyclin-dependent kinase and activator of Rb) and p19^Arf^ expression, improving the stability of p53 protein [165]. Therefore, it slows down the progression of cell cycle and induces senescence [166], leading to reduced adult NSC functioning and neurogenesis. However, deficiency of p16^INK4a^ in aged mice diminished this effect [167]. Let-7b initiates differentiation and inhibits proliferation of neural stem cells by targeting Tlx and cyclin D1 in adult NSC and embryonic brains [168]. A higher-to --lower/quiescent shift in NSC proliferative state from fetus to adult is contributed to by Imp1, a different let-7b target, even though it is not expressed in adult NSC [169]. As a result, changes in let-7b may initiate aging in adult NSC. The gene regulation mediated by micro RNAs impacts healthy aging as well as aging associated with neurodegenerative diseases [170]. Administering exosomes derived from NSCs (exo-NSCs) could restore BDNF signaling and memory in HFD mice [171], providing suggestive evidence of the potential therapeutic effect of exo-NSCs on HFD-induced NSC dysregulation in obesity. Hence, further studies on differential expression of certain exosomal non-coding RNAs must be performed to form an understandable association with pathological and healthy aging.

## 10. Challenges Associated with NSCs for Regenerative Medicine and Future Perspectives

Immunological rejection is one of the major difficulties in stem cell therapy. This could be addressed by isolating NSCs from the same subjects that require the therapy to prevent immunological reaction to the newly transplanted stem cells. Administering immunosuppressive drugs could be an additional or alternative option even though it has a lot of side effects. Another challenge is to make certain that the transplanted cells grow enough without causing tumor development and karyotypic instability. According to [172], there is a critical challenge in isolating multipotent NSCs from cell culture for transplantation as the majority of neurospheres in vitro are heterogenic with varying developmental stages and gene expression mainly due to ex vivo culture conditions. Overcoming these challenges and establishing NSC-based therapy for obesity-induced comorbidities, especially cardiovascular conditions, to improve functional outcomes through associated multimodal mechanisms is tremendously foresighted. Based on the difficulty in accessing the brain to collect tissues for processing from live animals, using induced pluripotent stem cell (iPSC) technology is a solution that could produce in vitro NSCs or neurons for transplantation. As iPSCs can be non-invasively obtained from live subjects, and to reduce the risk of immune rejection, reprogramming these cells to NSCs or neurons could provide autologous engraftments [173].

## 11. Conclusions

HtNSCs could be a potential therapy in obesity-induced cardiovascular diseases. Exosomes derived from htNSCs could be an alternative to or a conjunction with NSC therapy, being a minimally invasive technique to reverse aging and degenerative changes in the CNS. The relationship between htNSC dysregulation and sympathetic nerve response in obesity has never been studied. As brain microglia activation is a predominant indicator of neuroinflammation in hypertension, restoring a normal population of glia and neurons within the cardiogenic centers of the brain cannot be ruled out. Thus, identifying associated htNSC mechanisms and pathways could bring novel insight to therapeutic strategies in obesity-associated hypertension or sympathetic nerve overactivity.

## Figures and Tables

**Figure 1 cells-12-00769-f001:**
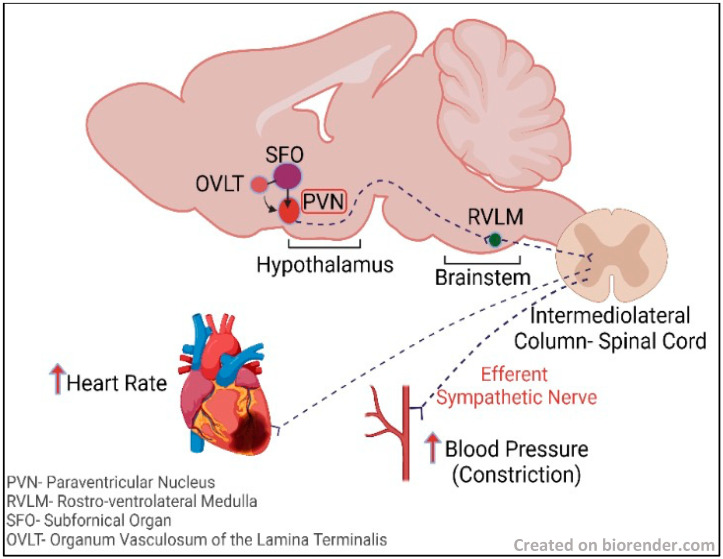
A simplified pathway showing excitatory circulatory signaling from SFO and OVLT, which are circumventricular organs lacking blood brain barrier, passing to the adjacent PVN in the hypothalamus and RVLM in the brainstem, causing an efferent sympathetic nerve response from the intermediolateral column of the spinal cord to the heart and blood vessels, resulting in increased heart rate and increased blood pressure associated with vasoconstriction.

**Figure 2 cells-12-00769-f002:**
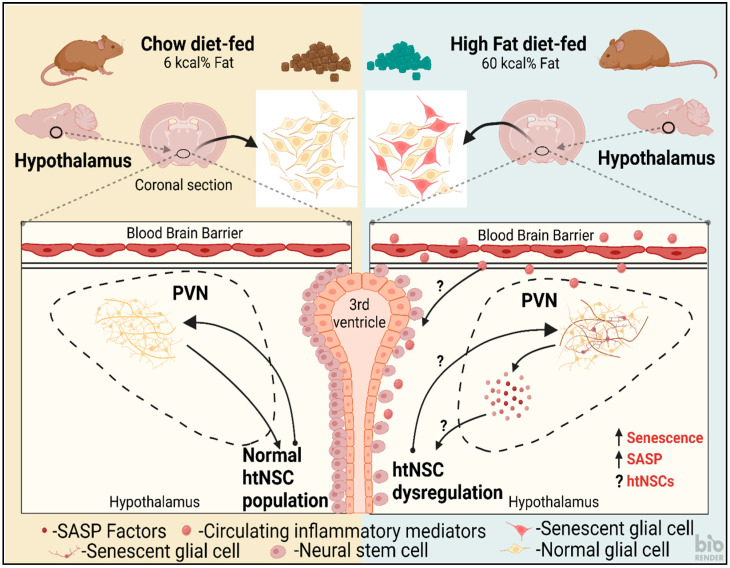
A schematic illustration of the response to feeding chow (left) and high fat diet (HFD) (right) on hypothalamic paraventricular nucleus (PVN) and hypothalamic neural stem cells (htNSCs) lining the 3rd ventricle in mice. HFD could potentially cause an increase in senescent glial cells within the PVN that can release senescence-associated secretory phenotype (SASP) factors causing a proinflammatory response, further leading to htNSC dysregulation. This could, again, cause a reduction in functional glia and neurons in the PVN. The circulatory inflammatory mediators could also potentially cross the blood brain barrier and cause a direct effect on the htNSCs. Chow-fed control showed normal cell populations.

**Figure 3 cells-12-00769-f003:**
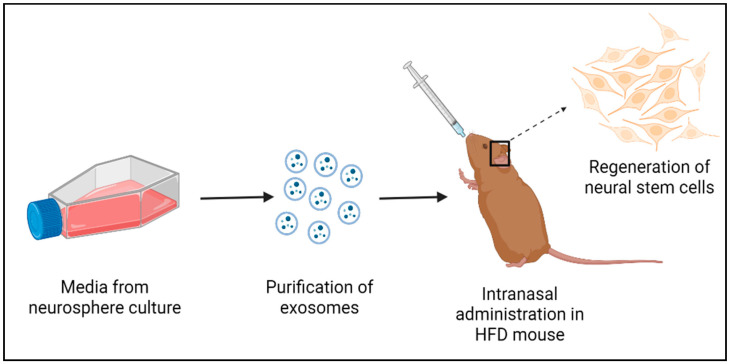
A simplified protocol for the potential therapeutic effect of exo-NSCs in obesity-induced NSC dysregulation.

## Data Availability

No new data was created as this is a review, and some data is unavailable due to privacy or ethical restrictions. Data will be published with ethical considerations in future manuscript.

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
