# Peer review of "Implications of Hypothalamic Neural Stem Cells on Aging and Obesity-Associated Cardiovascular Diseases"

_cells, 2023, doi:10.3390/cells12050769_

Round 1

Reviewer 1 Report

This is an interesting review centered on hypothalamic neurogenesis and its association with obesity and aging. The review is well written and organized, and contains a comprehensive amount of literature of great importance to those interested in neuroscience and the control of obesity and associated cardiovascular diseases.

The manuscript should be carefully revised for grammatical mistakes and some phrases that could be modified to more clearly display the information (some examples below).

Line 61. .....many neural progenitors

Line 90. Microglia are brain resident.........

Line 180. The last phrase starting in line 180 appears to have a contradictory statement as indicates that......the topmost contributors to obesity-induced anxiety are senescent cells and senolytic drugs.......

Line 272. Perhaps the authors meant to say "......it was concluded that maternal diabetes and differential exposure of the fetus to insulin and leptin could result in reduced growth or macrosomia that could have a significant effect......" ?

Line 277. Revise the first paragraph.

Line 344. Description of the abbreviation AL (as in AL diet) appears to be missing.

Author Response

Reviewer 1:
a) Grammatical mistakes and some phrases that could be modified to display the Information more clearly
Line 61. .....many neural progenitors
Many neural progenitors or specialized ependymal cells that are lining the 3rd ventricle are observed as glia-like tanycytes. They send processes to the arcuate nucleus and ventromedial nucleus of the hypothalamus.
b) Line 90. Microglia are brain resident.........
Microglia are brain resident macrophages that contribute to reduced neurogenesis in aging and play a predominant role in the inflammatory response.
c) Line 180. The last phrase starting in line 180 appears to have a contradictory statement as indicates that......the topmost contributors to obesity-induced anxiety are senescent cells and senolytic drugs.......
Hence by subsequent studies, they concluded that the topmost contributors to obesity-induced anxiety are senescent cells. Therefore, senolytic drugs have opened a novel therapeutic pathway to treat neuropsychiatric disorders.
d) Line 272. Perhaps the authors meant to say "......it was concluded that maternal diabetes and differential exposure of the fetus to insulin and leptin could result in reduced growth or macrosomia that could have a significant effect......" ? &
Line 277. Revise the first paragraph.
According to their studies, it was concluded that maternal diabetes and differential exposure of the fetus to insulin and leptin could result in reduced growth or macrosomia that could have a significant effect on the development of a fetal brain.
e) Line 344. Description of the abbreviation AL (as in AL diet) appears to be missing.
ad libitum (AL) diet.

Reviewer 2 Report

The manuscript describes the physiology and potential roles of hypothalamic neurogenesis and hypothalamic neural stem cells (htNSC) in the feeding behavior and cardiovascular function. Also, authors provide information obtained in other types and niches of neural stem cells (NSCs) and visualize the way this knowledge could be used to understand the htNSCs in physiological and pathophysiological conditions. Authors further describe how aging and neuroinflammation could be important therapeutic targets to restore the physiology of htNSCs and thus restore the hypothalamic functions.

The role of hypothalamic neurogenesis in feeding behavior and cardiovascular function is a relevant topic that must be further explored. However, the manuscript can be improved.

1.       Authors shall carefully address this topic specially in adult humans where neurogenesis is broadly debated even in well-studied regions such as the dentate gyrus (doi: 10.1126/science.abn8861).

Recent studies in human amygdala identified immature neurons that persist for many years without the presence of proliferative niches (doi: 10.1038/s41467-019-10765-1). The possibility of finding immature hypothalamic neurons instead of active proliferative niches that give rise to neurons in the adult human hypothalamus should be discussed.

2.       Section 10. Challenges associated with NSCs for regenerative medicine and future perspective:

Authors should also propose the use of induced pluripotent stem cells (iPSCs) in order to yield either NSCs or neurons for transplantation. Since iPSCs can be obtained from the patient by cell reprogramming (doi: 10.1038/nrm3448), NSCs or neurons derived from iPSCs could provide autologous engraftments that could reduce the risks of immune rejection.

Minor concern:

There are many sentences cited as “unpublished data”. Is it possible to find similar results within the literature so authors could cite them?

Author Response

Reviewer 2:
a) Authors shall carefully address this topic especially in adult humans where neurogenesis is broadly debated even in well-studied regions such as the dentate gyrus (doi: 10.1126/science.abn8861) & Recent studies in human amygdala identified immature neurons that persist for many years without the presence of proliferative niches (doi: 10.1038/s41467-019-10765-1). The possibility of finding immature hypothalamic neurons instead of active proliferative niches that give rise to neurons in the adult human hypothalamus should be discussed.
Line 74-80: However, a few studies showed the rare occurrences of proliferating neurogenic progenitors in the human dentate gyrus [32, 33] Also, one of the studies observed human paralaminar nuclei of the amygdala showing persistence of immature excitatory neurons for decades [34]. Thus, the possibility of observing immature non-proliferative hypothalamic neurons cannot be denied and future studies focusing on confirming their ability to proliferate and differentiate could possibly reveal their normal functionality.

b) Section 10. Challenges associated with NSCs for regenerative medicine and future perspective:
Authors should also propose the use of induced pluripotent stem cells (iPSCs) in order to yield either NSCs or neurons for transplantation. Since iPSCs can be obtained from the patient by cell reprogramming (doi: 10.1038/nrm3448), NSCs or neurons derived from iPSCs could provide autologous engraftments that could reduce the risks of immune rejection.
Lines 444-449:
Based on the difficulty in accessing the brain to collect tissues for processing from live animals, using induced pluripotent stem cell (iPSC) technology will be a solution to produce in vitro NSCs or neurons for transplantation. As iPSCs can be non-invasively obtained from live subjects and to reduce the risk of immune rejection, reprogramming these cells to NSCs or neurons could provide autologous engraftments [173].

Minor concern:
There are many sentences cited as “unpublished data”. Is it possible to find similar results within the literature so authors could cite them?
We will be submitting an article with all the unpublished data stated in the review as soon as possible.

Round 2

Reviewer 2 Report

The manuscript has been significantly improved and now it can be accepted.